# Detecting Adversarial Examples Is (Nearly) As Hard As Classifying Them

Florian Tramèr [1]

## Abstract

Making classifiers robust to adversarial examples is hard. Thus, many defenses tackle the seemingly easier task of *detecting* perturbed inputs.

We show a barrier towards this goal. We prove a general *hardness reduction* between detection and classification of adversarial examples: given a robust detector for attacks at distance $\epsilon$ (in some metric), we can build a similarly robust (but inefficient) *classifier* for attacks at distance $\epsilon/2$.

Our reduction is computationally inefficient, and thus cannot be used to build practical classifiers. Instead, it is a useful sanity check to test whether empirical detection results imply something much stronger than the authors presumably anticipated.

To illustrate, we revisit 13 detector defenses. For 11/13 cases, we show that the claimed detection results would imply an inefficient classifier with robustness far beyond the state-of-the-art.

## 1. Introduction

Consider the following claims about two defenses against adversarial examples (Szegedy et al., 2014) on CIFAR-10:

- defense A is a classifier that achieves robust accuracy of 90% under $\ell_\infty$-perturbations bounded by $\epsilon = {}^4\!/\!{}_{255}$;

- defense B also has a "detection" option, and achieves robust accuracy of 90% under $\ell_\infty$-perturbations bounded by $\epsilon = {}^8\!/\!{}_{255}$ (defense B is correct if it classifies a natural example correctly, and either detects or correctly classifies all perturbed examples at distance $\epsilon$.)

*If you had to take a bet that one of these two (empirical) claims is correct, which one would you choose?*

Defense A claims much higher robustness than the current best result achieved with adversarial training (Madry et al.,

2018), the only empirical defense against adversarial examples that stands the test of time.[1] Thus, this claim might be met with some initial skepticism and heightened scrutiny.

The claim of defense B is harder to assess, due to a lack of long-standing baselines for robust detectors. On one hand, detection of adversarial examples has largely been considered to be an easier task than classification (Xu et al., 2018; Pang et al., 2021). On the other hand, defense B claims robustness to perturbations that are twice as large as defense A.

In this paper, we show that the claims about defenses A and B are, in fact, equivalent! (up to computational efficiency.)

We prove a general *hardness reduction* between classification and detection of adversarial examples. Given a detector defense that achieves robust risk $\alpha$ for attacks at distance $\epsilon$ (under any metric), we show how to build an *explicit but inefficient* classifier that achieves robust risk $\alpha$ for classifying attacks at distance $\epsilon/2$. The reverse implication also holds: a classifier robust at distance $\epsilon/2$ implies an explicit but inefficient robust detector at distance $\epsilon$.

To the authors knowledge, there is no known way of leveraging computational *inefficiency* to build more robust models. We should thus be as "surprised" by the claim made by defense B as by the claim made by defense A.

Our hardness reduction provides a way of assessing the plausibility of new robust detection claims, by contrasting them with results from the more mature literature on robust classification. To illustrate, we revisit 13 published detection defenses, and show that in 11/13 cases the defense's robust detection claims would imply an inefficient classifier with robustness far superior to the current state-of-the-art. Yet, none of these defense papers claim that their results should imply such a breakthrough in robust *classification*.

Using our reduction, it is obvious that most of the 13 detection defenses are claiming stronger robustness than we currently believe feasible. And indeed, many of these defenses were later shown to have overestimated their robustness (Carlini & Wagner, 2017; Tramèr et al., 2020).

Remarkably, we find that for *certified* defenses, the state-

---

[1]Stanford University. Correspondence to: Florian Tramèr <tramer@cs.stanford.edu>.

*Accepted by the ICML 2021 workshop on A Blessing in Disguise: The Prospects and Perils of Adversarial Machine Learning.* 

---

[1]The current state-of-the-art $\ell_\infty$ robustness for $\epsilon = {}^4\!/\!{}_{255}$ on CIFAR-10 (without external data) is $\approx 79\%$ (Rebuffi et al., 2021).

of-the-art results for provable robust classification and detection perfectly match the results implied by our reduction. For example, Sheikholeslami et al. (2021) recently proposed a certified detector on CIFAR-10 with provable robust error that is within $3\%$ of the provable error of the *inefficient* detector obtained by combining the state-of-the-art robust classifier of Zhang et al. (2020) and our result. This gives further credence to our assumption that computational inefficiency is of little help towards building more robust models.

In summary, we prove that giving classifiers access to a rejection/detection option does not help robustness. It would be interesting to find similar "barriers" for other directions that have been considered to enhance robustness (e.g., the use of randomness, ensembles, denoising functions, etc.)

## 2. Hardness Reductions Between Robust Classifiers and Detectors

In this section, we prove our main result: a reduction between robust detectors and robust classifiers, and vice-versa.

We consider a classification task with a distribution $\mathcal{D}$ over examples $x \in \mathbb{R}^d$ with labels $y \in [C]$. A classifier is a function $f : \mathbb{R}^d \to [C]$. A detector is a classifier with an extra "rejection" option $\perp$. The binary indicator function $\mathbb{1}_{\{A\}}$ is 1 if and only if the predicate $A$ is true.

We first define a classifier's *risk*, i.e., its classification error on unperturbed samples.

**Definition 1** (Risk). Let $f : \mathbb{R}^d \to [C] \cup \{\perp\}$ be a classifier (optionally with a detection output $\perp$). The risk of $f$ is:

$$R(f) := \mathop{\mathbb{E}}_{(x,y)\sim\mathcal{D}} \left[ \mathbb{1}_{\{f(x)\neq y\}} \right] \quad (1)$$

Note that for a detector, rejecting an unperturbed example, $f(x) = \perp$, is counted as an error.

For classifiers without a rejection option, we define the *robust risk* as the risk on worst-case adversarial examples. Given an input $x$ sampled from $\mathcal{D}$, an adversarial example $\hat{x}$ is constrained to being within distance $d(x, \hat{x}) \leq \epsilon$ from $x$, where $d$ is some arbitrary distance measure.

**Definition 2** (Robust risk). Let $f : \mathbb{R}^d \to [C]$ be a classifier. The robust risk at distance $\epsilon$ is:

$$R_{\text{adv}}^{\epsilon}(f) := \mathop{\mathbb{E}}_{(x,y)\sim\mathcal{D}} \left[ \max_{d(x,\hat{x})\leq\epsilon} \mathbb{1}_{\{f(\hat{x})\neq y\}} \right] \quad (2)$$

For a detector (a classifier with an extra rejection output), we analogously define the robust risk with detection. The classifier is now allowed to reject adversarial examples.

**Definition 3** (Robust risk with detection). Let $f : \mathbb{R}^d \to [C] \cup \{\perp\}$ be a classifier with an extra detection output $\perp$.

The robust risk with detection at distance $\epsilon$ is:

$$R_{\text{adv-det}}^{\epsilon}(f) := \mathop{\mathbb{E}}_{(x,y)\sim\mathcal{D}} \left[ \max_{d(x,\hat{x})\leq\epsilon} \mathbb{1}_{\{f(x)\neq y \,\vee\, f(\hat{x})\notin\{y,\perp\}\}} \right] \quad (3)$$

That is, a detector defense $f$ is robust on a natural input $x$ if and only if $f$ classifies $x$ correctly and $f$ either rejects or correctly classifies every input $\hat{x}$ within distance $\epsilon$ from $x$. The requirement that $f$ correctly classify natural examples eliminates pathological defenses that reject all inputs.

We are now ready to introduce our main result, a reduction from a robust detector for adversarial examples at distance $\epsilon$, to a robust classifier at distance $\epsilon/2$:

**Theorem 4** ($\epsilon$-detection implies $\epsilon/2$-classification). *Let $d(\cdot,\cdot)$ be an arbitrary metric. Let $f$ be a defense that achieves risk $R(f) = \alpha$, and robust risk with detection $R_{\text{adv-det}}^{\epsilon}(f) = \beta$. Then, we can construct an explicit (but inefficient) defense $g$ that achieves risk $R(g) \leq \alpha$ and robust risk $R_{\text{adv}}^{\epsilon/2}(g) \leq \beta$.*

*The defense $g$ is constructed as follows on input $x$:*

- *Run $y = f(x)$. If $y \neq \perp$, output $y$.*
- *Otherwise, find an input $x'$ such that $d(x, x') \leq \epsilon/2$ and $f(x') \neq \perp$. If such an input $x'$ is found, output $f(x')$. Else, output a random class $y \in [C]$.*

An intuitive illustration for our construction, and for the proof of the theorem (see below) is in Figure 1.

Our construction is best viewed as an analog of *minimum distance decoding* in coding theory. We can view the clean data sampled from $\mathcal{D}$ as codewords, and adversarial examples $\hat{x}$ as a noisy message with a certain number of errors (where the error magnitude is measured using an arbitrary metric on $\mathbb{R}^d$ rather than the Hamming distance). A standard result in coding theory states that if a code can *detect* $\alpha$ errors, then it can correct $\alpha/2$ errors.

*Proof.* First, note that the natural accuracy of $g$ is at least as high as that of $f$, since $g$ always mimics the output of $f$ when $f$ does not reject an input. Thus, $R(g) \leq R(f) = \alpha$.

Now, consider an input $(x,y) \sim \mathcal{D}$ for which $g$ is not robust at distance $\epsilon/2$. That is, there exists an input $\hat{x}$ at distance $d(x, \hat{x}) \leq \epsilon/2$ such that $g(\hat{x}) = \hat{y} \neq y$. We will show that the defense $f$ is not robust with detection for $x$ either (for attacks at distance up to $\epsilon$.)

By definition of $g$, if $g(\hat{x}) = \hat{y} \neq y$ then either:

- The defense $f$ also misclassifies $\hat{x}$, i.e., $f(\hat{x}) = \hat{y}$.
  So $f$ is not robust with detection for $x$ at distance $\epsilon$.
- There exists $x'$ such that $d(\hat{x}, x') \leq \epsilon/2$ and $f(x') = \hat{y}$.
  Note that by the triangular inequality, $d(x, x') \leq d(x, \hat{x}) + d(\hat{x}, x') \leq \epsilon/2 + \epsilon/2 = \epsilon$, and thus $f$ is not robust with detection for $x$ at distance $\epsilon$.

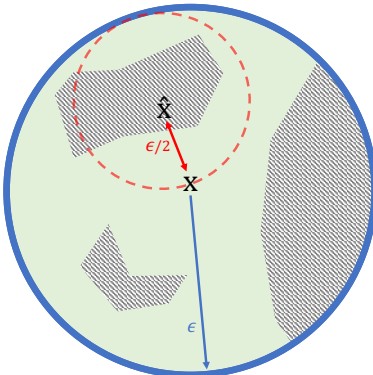

*Figure 1.* Illustration of the construction of a robust classifier from a robust detector in Theorem 4. The outer blue circle represents all inputs at distance at most $\epsilon$ from the input $x$. For a detector $f$, the areas in green correspond to correctly classified inputs, and ratcheted gray areas correspond to rejected inputs. The detector $f$ is thus robust on $x$ up to distance $\epsilon$. The classifier $g$ classifies a perturbed input $\hat{x}$, at distance $\epsilon/2$ from $x$, by finding any input within distance $\epsilon/2$ from $\hat{x}$ (the red dashed circle) that is not rejected by $f$. The classifier $g$ is robust on $x$ up to distance $\epsilon/2$.

- For all $x'$ such that $d(\hat{x}, x') \leq \epsilon/2$, we have $f(x') = \perp$ (and thus $g$ has output $\hat{x}$ at random).

  Since $d(x, \hat{x}) \leq \epsilon/2$, this implies that $f(x) = \perp$, and thus $f$ is not robust with detection for $x$.

In summary, whenever $g$ fails to robustly classify an input $x$ up to distance $\epsilon/2$, the defense $f$ also fails to robustly classify $x$ with detection up to distance $\epsilon$. Taking expectations over the entire distribution $\mathcal{D}$ concludes the proof. □

A corollary to our reduction is that many "information theoretic" results about robust classifiers can be directly extended to robust detectors. For example, the formal tradeoff between robust classification and accuracy of Tsipras et al. (2019), the increased data complexity of robust generalization of Schmidt et al. (2018), or the tradeoff between robustness to multiple perturbation types of Tramèr & Boneh (2019), all imply similar tradeoffs for robust detectors. Indeed, all of these results also apply to inefficient classifiers.

A similar argument can be used in the opposite direction, to show that a robust classifier at distance $\epsilon/2$ implies an inefficient robust detector at distance $\epsilon$.

**Theorem 5** ($\epsilon/2$-classification implies $\epsilon$-detection). *Let $d(\cdot, \cdot)$ be an arbitrary metric. Let $g$ be a defense that achieves robust risk $R^{\epsilon/2}_{adv}(f) = \beta$. Then, we can construct an explicit (but inefficient) defense $f$ that achieves risk $R(f) \leq \beta$ and robust risk with detection $R^{\epsilon}_{adv\text{-}det}(f) \leq \beta$.*

*The defense $f$ is constructed as follows on input $x$:*

- *Run $y = g(x)$.*
- *Find an input $x'$ such that $d(x, x') \leq \epsilon/2$ and $g(x') \neq y$.*

*If such an input $x'$ exists, output $\perp$. Else, output $y$.*

The proof of Theorem 5 is in Appendix A.

A main distinction between Theorem 4 and Theorem 5 is that the construction in Theorem 4 preserves clean accuracy, but the construction in Theorem 5 does not. That is, the constructed robust detector in Theorem 5 has natural accuracy that is equal to the robust classifier's robust accuracy.

The construction in Theorem 5 can be efficiently (but approximately) instantiated by a *certifiably robust* classifier (Wong & Kolter, 2018; Raghunathan et al., 2018). These defenses can certify that a classifier's output is constant for all points within some distance of the input. For an adversarial example $\hat{x}$ for $g$, the certification always fails and thus the constructed detector $f$ will reject $\hat{x}$. If $g$ is robust and the certification succeeds, the detector $f$ copies the output of $g$. However, a certified defense may fail to certify a robust input (a false negative), and thus the detector $f$ may reject more inputs than with the "optimal" construction in Theorem 5. This reduction from a certified classifier to a detector is implicit in (Wong & Kolter, 2018, Section 3.1).

## 3. What Are Detection Defenses Claiming?

We now survey 13 detection defenses, and consider the robust *classification* performance that these defenses implicitly claim (via Theorem 4). As we will see, in 11/13 cases, the defenses' detection results imply an inefficient classifier with far better robust accuracy than the state-of-the-art.

These claims are *not necessarily wrong*. But given how challenging robust classification is proving to be, we have reason to be skeptical of major breakthroughs (even for inefficient classifiers). To compound this, many proposed detection defenses are quite simple, and reject adversarial inputs based on some standard statistical test over a neural network's features. It would be particularly surprising if such simple techniques could yield robust *classifiers* (and indeed, many of these defenses have been broken by stronger attacks (Carlini & Wagner, 2017; Tramèr et al., 2020)).

**Setup.** We choose 13 detector defenses from the literature (see Table 1). Our choice for these defenses was somewhat artificial and pragmatic: we chose defenses that made claims that were easy to translate into a bound on the robust risk with detection $R^{\epsilon}_{adv\text{-}det}$. Indeed, some detection defenses simply report a single AUC score, from which we cannot derive a useful bound on the robust risk. We thus focus on defenses that either directly report a robust error akin to Definition 3, or that provide concrete pairs of false-positive and false-negative rates (e.g., a full ROC curve). In the latter case, we compute a "best-effort" bound on the robust risk

*Table 1.* For each detector defense, we compute a (best-effort) bound on the claimed robust risk with detection $R^\epsilon_{\text{adv-det}}$, and report the complement (the robust accuracy with detection). For each detector's robustness claim (at distance $\epsilon$), we report the state-of-the-art robust classification accuracy $1 - R^{\epsilon/2}_{\text{adv}}$, for attacks at distance $\epsilon/2$. Detection defense claims that imply a higher robust classification accuracy than the state-of-the-art are in red.

| Dataset | Defense | Norm | $\epsilon$ | $1 - R^\epsilon_{\text{adv-det}}$ | $1 - R^{\epsilon/2}_{\text{adv}}$ |
|---------|---------|------|------------|-----------------------------------|-----------------------------------|
| MNIST | Grosse et al. (2017) | $\ell_\infty$ | 0.5 | $\geq 98\%$ | 94% |
| | Ma et al. (2018) | $\ell_2$ | 4.2 | $\geq 99\%$ | 72% |
| CIFAR-10 | Yin et al. (2020) | $\ell_2$ | 1.7 | $\geq 90\%$ | 66% |
| | Feinman et al. (2017) | $\ell_2$ | 2.7 | $\geq 43\%$ | 36% |
| | Miller et al. (2019) | $\ell_2$ | 2.9 | $\geq 75\%$ | 30% |
| | Ma & Liu (2019) | $\ell_\infty$ | $4/255$ | $\geq 96\%$ | 85% |
| | Roth et al. (2019) | $\ell_\infty$ | $8/255$ | $\geq 66\%$ | 79% |
| | Lee et al. (2018) | $\ell_\infty$ | $20/255$ | $\geq 81\%$ | 59% |
| | Li et al. (2019) | $\ell_\infty$ | $26/255$ | $\geq 80\%$ | 44% |
| ImageNet | Xu et al. (2018) | $\ell_2$ | 1.0 | $\geq 67\%$ | 54% |
| | Ma & Liu (2019) | $\ell_\infty$ | $2/255$ | $\geq 68\%$ | 55% |
| | Jha et al. (2019) | $\ell_\infty$ | $2/255$ | $\geq 30\%$ | 55% |
| | Hendrycks & Gimpel (2017) | $\ell_\infty$ | $10/255$ | $\geq 76\%$ | 30% |
| | Yu et al. (2019) | $\ell_\infty$ | $26/255$ | $\geq 7\%$ | 5% |

*Table 2.* Certified robust accuracy $1 - R^{\epsilon/2}_{\text{adv}}$ for the defense of Zhang et al. (2020), and certified robust accuracy with detection $1 - R^\epsilon_{\text{adv-det}}$ for the defense of Sheikholeslami et al. (2021).

| $\epsilon$ | $1 - R^\epsilon_{\text{adv-det}}$ | $1 - R^{\epsilon/2}_{\text{adv}}$ |
|------------|-----------------------------------|-----------------------------------|
| $8/255$ | 37% | 39% |
| $16/255$ | 32% | 33% |

with detection as:

$$R^\epsilon_{\text{adv-det}}(f) \leq \text{FPR} + \text{FNR} + R(f) \,,$$

where FPR and FNR are the detector's false-positive and false-negative rates, and $R(f)$ is the defense's standard risk (i.e., the test error on natural examples). We note that this union bound may be quite pessimistic, as we might over-count examples that lead to multiple sources of errors (e.g., a natural input that is misclassified and erroneously detected). The true robustness claim made by these detector defenses might thus be stronger than what we obtain from our bound. We encourage future defense papers to report the adversarial risk, to facilitate comparisons with robust classifiers.

The 13 detector defenses use three datasets: MNIST, CIFAR-10 and ImageNet, and consider adversarial examples under the $\ell_\infty$ or $\ell_2$ norms. Given a claim of robust detection at distance $\epsilon$, we contrast it to a state-of-the-art robust classification result for distance $\epsilon/2$. On MNIST with $\ell_\infty$ attacks, we use TRADES (Zhang et al., 2019) and measure robust error with the Square attack (Andriushchenko et al., 2020). For $\ell_2$ attacks, we use the model from Tramèr & Boneh (2019) and measure robust error with PGD (Madry et al., 2018). For CIFAR-10, we use the best model of Rebuffi et al. (2021) (trained without external data), and attack it using AutoAttack (Croce & Hein, 2020). For ImageNet, we use models and attacks from Engstrom et al. (2019).

We also consider two *certified* defenses for CIFAR-10: the robust classifier of Zhang et al. (2020), and a recent certified detector of Sheikholeslami et al. (2021).

**Results.** As we can see from table 1, most defenses claim a detection performance that implies a far greater robust accuracy than our current best robust classifiers.

In Table 2, we look at the robust accuracy with detection, and standard robust accuracy achieved by *certified* defenses (for which the claimed robustness numbers are necessarily mathematically correct). Remarkably, we find that existing results nearly match what is implied by our reduction (up to $\pm 2\%$ error). For example, Zhang et al. (2020) follow a long line of results on robust classifiers and achieve 39% robust accuracy on CIFAR-10 for perturbations of $\ell_\infty$-norm below $4/255$. Together with Theorem 5, this implies an inefficient detector with 39% robust detection accuracy for perturbations of $\ell_\infty$-norm below $8/255$. The recent work of Sheikholeslami et al. (2021) nearly matches that bound (37% robust accuracy with detection), with a defense that has the advantage of being concretely efficient.

## 4. Conclusion

We have shown formal reductions between robust classification with, and without, a detection option. Our results show that significant progress on one of these two tasks implies similar progress on the other. This raises the question on whether we should spend our efforts on studying both of these tasks, or focus our efforts on a single one.

On one hand, the two tasks represent different ways of tackling a same goal, and working on either task might result in new techniques or ideas that apply to the other task as well. On the other hand, our reductions show that unless we make progress on both tasks, work on one of the tasks can merely aim to match the robustness of our inefficient constructions, whilst improving their computational complexity.

## Acknowledgments

The author would like to thank Nicholas Carlini and Wieland Brendel for helpful discussions, as well as Alex Ozdemir for suggesting the connection between our results and minimum distance decoding.

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

# A. Proof of Theorem 5.

We restate Theorem 5 here:

**Theorem 4** ($\epsilon/2$-classification implies $\epsilon$-detection)**.** *Let $d(\cdot, \cdot)$ be an arbitrary metric. Let $g$ be a defense that achieves robust risk $R_{adv}^{\epsilon/2}(f) = \beta$. Then, we can construct an explicit (but inefficient) defense $f$ that achieves risk $R(f) \leq \beta$ and robust risk with detection $R_{adv\text{-}det}(f; \epsilon) \leq \beta$.*

*The defense $f$ is constructed as follows on input $x$:*

- *Run $y = g(x)$.*
- *Find an input $x'$ such that $d(x, x') \leq \epsilon/2$ and $g(x') \neq y$. If such an input $x'$ exists, output $\perp$. Else, output $y$.*

*Proof of Theorem 5.* Note that for any input $(x, y)$ for which $g$ is robust at distance $\epsilon/2$, no input $x'$ above exists and so $f(x) = y$. Thus, the risk of $f$ is at most $\beta$.

Now, consider an input $(x, y) \sim \mathcal{D}$ for which $f$ is not robust with detection at distance $\epsilon$. That is, either $f(x) \neq y$, or there exists an input $\hat{x}$ at distance $d(x, \hat{x}) \leq \epsilon$ such that $f(\hat{x}) = \hat{y} \notin \{y, \perp\}$. We will show that the defense $g$ is not robust for $x$ either (for attacks at distance up to $\epsilon/2$.)

If $f(x) \neq y$, then by the same argument as above it cannot be the case that $g$ is robust at distance $\epsilon/2$ for $x$.

So let us consider the case where $f(\hat{x}) = \hat{y} \notin \{y, \perp\}$. By the definition of $f$, this means that for all $x'$ at distance at most $\epsilon/2$ from $\hat{x}$, we have $g(x') = \hat{y}$. But, note that there exists a point $x^*$ that is at distance at most $\epsilon/2$ from both $\hat{x}$ and $x$. Since we must have $g(x^*) = \hat{y}$, we conclude that $g$ is not robust at distance $\epsilon/2$ for $x$.

Taking expectations over the distribution $\mathcal{D}$ concludes the proof. □