# OpenReview forum: "Detecting Adversarial Examples Is (Nearly) As Hard As Classifying Them"
_ICML.cc/2021/Workshop/AML — ICML 2021 Workshop AML Oral_

### Official Review · Reviewer_pfiE · 2021-06-19
**Interesting work**

**Rating:** Accept
**Confidence:** 5

**Review:**

This is good work connecting the areas of robust classifying and robust detection, especially in the cases that the evaluations against detection methods are usually under-explored, compared to those against classifying methods. The construction of equivalent classifiers and detectors in Theorem 4 and 5 are elegant.

Connecting classification with detection/rejection modules is an intriguing topic. A concurrent work [1] also compares the learning difficulties between a robust classifier and a robust detector (w.r.t. number of classes), where an easier-to-learned rejection module can be coupled with the classifier.

I think the conclusions in the submitted paper can provide beneficial insights for the community, and avoid overclaims in future detection papers.

[1] Adversarial Training with Rectified Rejection. arXiv 2105.14785

---

### Decision · Program_Chairs · 2021-06-21

**Decision:**

Accept (Oral)

**Comment:**

A good work considering robust classification and robust detection, as agreed by the reviewer. This paper could be significant to the workshop.